# Effective Field Collection of Pezizales Ascospores for Procuring Diverse Fungal Isolates

Alassane Sow [1,†], Judson Van Wyk [2,†], Benjamin Lemmond [3], Rosanne Healy [3], Matthew E. Smith [3] and Gregory Bonito [1,2,*]

1. Department of Microbiology, Genetics, and Immunology, Michigan State University, East Lansing, MI 48824, USA; sowalassa@outlook.com
2. Department of Plant, Soil and Microbial Sciences, Michigan State University, East Lansing, MI 48824, USA; vanwykju@msu.edu
3. Department of Plant Pathology, University of Florida, Gainesville, FL 32611, USA; benlemmond@gmail.com (B.L.); rosanne.healy@gmail.com (R.H.)
* Correspondence: bonito@msu.edu
† These authors contributed equally to this work.

**Abstract:** Pezizales are a diverse and economically important order of fungi. They are common in the environment, having epigeous form, such as morels and hypogeous, forms called truffles. The mature ascospores of most epigeous Pezizales are forcibly discharged through an opening at the ascus apex created with the lifting of the operculum, a lid-like structure specific to Pezizales. The axenic cultures of Pezizales fungi isolated from single ascospores are important for understanding the life cycle, development, ecology, and evolution of these fungi. However, obtaining single-spore isolates can be challenging, particularly for collections obtained in locations where sterile work environments are not available. In this paper, we introduce an accessible method for harvesting ascospores from fresh ascomata in the field and laboratory for obtaining single-spore isolates. Ascospores are harvested on the inside cover of Petri plate lids in the field, air dried, and stored. At a later date, single-spore isolates are axenically cultured through serial dilution and plating on antibiotic media. With this approach, we were able to harvest ascospores and obtain single-spore isolates from 12 saprotrophic and 2 ectomycorrhizal species belonging to six Pezizales families: Discinaceae, Morchellaceae, Pezizaceae, Pyronemataceae, Sarcosomataceae, and Sarcoscyphaceae. This method worked well for saprotrophic taxa (12 out of 19 species, 63%) and was even effective for a few ectomycorrhizal taxa (2 out of 13 species, 15%). This process was used to study the initial stages of spore germination and colony development in species across several Pezizales families. We found germination often commenced with the swelling of the spore, followed by the emergence of 1–8 germ tubes. This method is sufficiently straightforward that, provided with sterile Petri dishes, citizen scientists from distant locations could use this approach to capture spores and subsequently mail them with voucher specimens to a research laboratory for further study. The generated single-spore Pezizales isolates obtained through this method were used to generate high-quality genomic data. Isolates generated in this fashion can be used in manipulative experiments to better understand the biology, evolution, and ecogenomics of Pezizales.

**Keywords:** ascomycete; biodiversity; collection methods; axenic culture; ectomycorrhizal; ITS; LSU; saprotrophic

## 1. Introduction

Pezizales, the only order in the class Pezizomycetes, contains approximately 23 families, 200 genera, and 2000 species and exhibits a wide range of morphological and biological characteristics [1]. These fungi are ecologically important and diverse and include coprophiles, pyrophilous species, parasites of plants and fungi, saprotrophs, bacterial farmers, endophytes, and ectomycorrhizal symbionts [2–5]. However, Pezizales diversity is still



underdescribed, the biology and ecology of many taxa are not well understood, and the phylogenetic relationships among families are not well supported [2].

Fungal taxonomy and natural history are based upon voucher specimens from field collections that are curated and preserved in institutional natural science collections (e.g., fungaria), where they are accessible to other researchers. These vouchered specimens enable phylogenetic research, provide species-concept hypotheses through the study of morphological characteristics and type collections, and can serve future generations of mycological inquiry in understanding the effect of global change on fungal diversity, phenology, and evolution. As critical as voucher specimens are in fungal research, they provide different information than that which can be obtained from living fungal isolates [6].

Research based on fungal isolates has increased the understanding of fungal ecology, physiology, developmental biology, and genomics. Furthermore, species of fungi are often described using their morphological characteristics during sexual or asexual reproduction. For example, asexual reproduction has long been documented and reported from culture collections where this process has been observed and studied in living fungi (e.g., [7–9]. Survival structures, such as chlamydospores or sclerotia, have been observed in the living cultures of some species of Pezizales, thereby helping to elucidate the lifecycles of these fungi [10]. Axenic cultures also allow for substrate utilization, metabolomics, and transcriptomic gene expression investigations [11–13].

Many species of Pezizales can be readily cultured through tissue propagation [14] or by collecting and germinating spores directly on media [15,16]. However, destructive sampling may be problematic when collections are of rare taxa, ascomata are tiny, or ascomata are few. This approach is most challenging in situations where there is no sterile environment to axenically culture specimens in the window of time that the ascomata are fresh. However, with the exception of *Glaziella* and some species of Pseudombrophilaceae, epigeous Pezizales are notable for their forcible discharge of ascospores from operculate asci. Given the synchronized discharge of ascospores in large numbers, spores can be collected from ascomata for culturing. Volk and Leonard [17] reported placing bisected *Morchella* ascomata onto Petri dishes to collect spores en masse for generating cultures. Karakehian et al. [18] presented step-by-step methods for culturing ascomycete fungi directly from ascospores by placing a section of a mature ascoma in an inverted Petri plate of nutrient media and allowing the ascospores to be ejected directly onto suitable growth media. While this method is useful for generating pure cultures, it still requires media and axenic conditions to be available when collections are fresh. We were interested in finding a simple method for spore capture in the field. Ideally, the method would be easy and could be used by other researchers, and even citizen scientists, interested in aiding fungal biology research.

Here, we describe a method for generating axenic, single-spore isolates from diverse Pezizales while maintaining the condition of specimens to ensure high-quality fungarium collections. Briefly, ascomata collected in the field are carefully placed in sterile Petri plates for several hours and allowed to discharge spores. The ascomata are then removed, dried, and vouchered, while the spores are stored in a clean, cool, dry environment until they can be cultured. This accessible method enables the collection of ascospores in the field, which can be used at a later date by researchers to generate cultures. To assess the utility of this method, we collected ascospores in the field from diverse Pezizales during field campaigns and tested whether we could generate single-spore isolates from these harvested ascospores. With this approach, we were successful in obtaining cultures from 15 different species of Pezizales spanning eight genera in six families, all while maintaining the value of the original collections. We expect this simple, low-tech method to enable further studies on the development, function, and physiology of Pezizales, and perhaps other fungi, as the approach is adaptable to other fungi that forcibly discharge their spores.

## 2. Materials and Methods

### 2.1. Collecting Pezizales

Collections of Pezizales ascomata were made in Michigan, USA, from spring 2020 to spring 2022, and during the fall of 2021 in the Southern Appalachian mountains (North Carolina and Tennessee, USA). To obtain ascospores from fresh specimens, collected ascomata were gently cleaned of loose soil and other environmental debris and were enclosed in a sterile 60 × 15 mm Petri plate. Kimwipes (Kimtech, Roswell, Georgia, USA), tape, or other materials were used to orient and stabilize ascomata so they were in an upright position, allowing them to forcefully eject their spores directly onto the inner surface of the Petri plate lid. In some cases, Pezizales cannot be removed from their substrate without destroying the collection. In such cases, ascomata should be raised as close to the surface of the Petri plate as possible by propping them up with a tissue or even a green leaf (see Figure 1A). This focuses the spore dispersal to a concentrated location on the Petri plate and makes it easier to locate the spores. The areas around the target ascoma can then be labeled, and any undesirable spores can be effectively avoided. In the absence of Petri plates, specimens were brought back to the laboratory and set up in a Petri plate the same day. Plates were taped shut so the lid would not fall off during transport, and materials were carefully transported in an upright position from the field to the laboratory in a tackle box. After two to five hours, each ascoma was removed from its Petri plate, carefully lifting the lid so that any spores discharged from the humidity and pressure change would be caught on the lid. For some taxa, spore discharge could be synchronized in this way. Ascospore collections were confirmed by visualizing the Petri plate lids under a microscope at 100× and 40× magnification (Figure 1). Dense clusters of spores were circled with a permanent marker so they could be processed quickly at a later date (Figure 1E). Ascomata were further identified based on morphological characteristics (e.g., [19–21]), ITS rDNA sequencing (described below), and then were dried and accessioned at either Michigan State University Herbarium (MSC) or University of Florida Fungarium (FLAS-F) for further study. Petri plate bases were cleaned with ethanol or replaced to prevent unwanted materials from contaminating the spore deposit. Plates were air-dried of residual moisture by taping the lid of each plate to its base in two places to hold the plate together and protect it from contamination without completely sealing it (Figure 1C,D). Once plates were completely dried (within 24 h of spore collection), they were Parafilmed closed, stacked in a plastic bag, and stored in a clean, dry place at room temperature (20 °C) for further culturing and microscopy.

### 2.2. Germinating Ascospores

Before attempting to germinate spores, collections of taxa with known ecologies were separated according to their ecological guild (e.g., saprotrophic or ectomycorrhizal). Spores from saprotrophic species were incubated on 1% malt extract agar (MEA), while ectomycorrhizal species or individuals with unknown ecologies were incubated on modified Melin–Norkrans (MMN) media [22]. Spores of all specimens were also plated on 10% water agar (WA). Saprotrophic fungi are known to generally grow well on MEA, whereas ectomycorrhizal fungi often need specific media and supplemented vitamins to grow well, such as MMN [22]. In some cases, bacteria are known to codisperse with some fungi spores. thereby making fungal isolation difficult [18]. To prevent bacterial growth, all of the media contained the antibiotics streptomycin (100 mg/L), rifampicin (50 mg/L), and chloramphenicol (100 mg/L).

Before plating in sterile conditions, spore prints were visualized under 40× magnification, and the area with the highest density of spores was circled with a permanent marker. Then, 100 µL of sterile deionized water was gently pipetted up and down multiple times within the marked area. Because spores are sometimes not visible to the naked eye, observations of 10 µL of each spore slurry was deposited on a slide and viewed with a compound microscope at 100× magnification to ensure target spores were suspended in the water. Ascospores can clump together in water, making it difficult to create a spore

suspension [23]. In such cases, spores were first disturbed with a pipette tip, retracted into the pipette tip, and then placed into another 100 µL of sterile deionized water inside of a sterile 1.5 mL tube. In exceptional cases, a dilute Tween 20 (0.01%) solution was used to reduce ascospore clumping during dilution plating [24]. To confirm spore presence and ascertain their concentration, 10–20 µL of the spore "slurry" was pipetted onto a hemocytometer and observed under 100× magnification. For single-spore isolates, the optimal spore concentration of ≤1 spore/µL was targeted so that germinated spores were sufficiently dispersed on the surface of the plate to allow for accurate visualization. Therefore, dilutions were sometimes necessary to reach this spore concentration. Before plating, spore solutions were incubated for one hour at room temperature to rehydrate the spores. Then, 100 µL of each mixture was pipetted onto 100 x 15mm plates of media and a sterile glass cell spreader was used to evenly spread the spores onto the media. After inoculation, each plate was observed under a compound microscope to confirm that a low density of spores was present on the plates. In addition, the locations of individual spores that were distant from others were circled on the lid with a permanent marker to aid subsequent observations and isolations. Each plate was wrapped with Parafilm (Amco, Zürich, Switzerland) and incubated in the dark at room temperature.

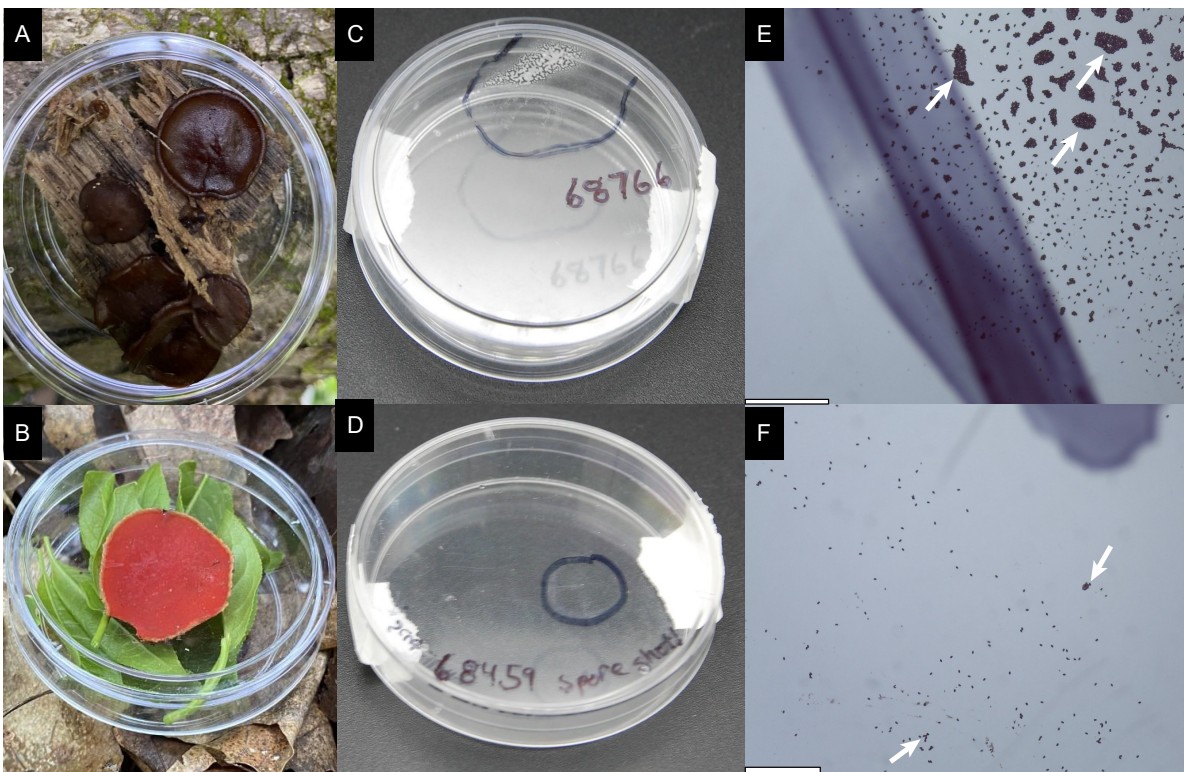

**Figure 1.** Pezizales collections in the field and confirmation of spore dispersal in the laboratory. (**A**) *Pachyella* (FLAS-F-69780) collection still attached to woody substrate in a 60 × 15 mm plate. (**B**) *Sarcoscypha* sp. propped up using living leaves in a 60 × 15 mm plate. (**C**) Spores of *Legliana* (FLAS-F-68766), where the spore print was circled with a marker; the line separates the spore print from the surface without spores. (**D**) Spores of *Scutellinia* sp. (FLAS-F-68459), where the spore print was circled with a marker. (**E**) Confirming presence of *Legliana* sp. (FLAS-F-68766) spores within the marked circle shown in (**C**); note that spores (dark clusters) are present inside the circled area but are not present outside of the border. (**F**) Confirming presence of *Scutellinia* sp. (FLAS-F-68459) spores (dark specks) within the marked circle as shown in (**D**) = 500 µm, (**F**) = 500 µm.

Plates were observed daily to monitor for germination and development. Germinated spores were photographed under a Lecia DM750 compound microscope (Lecia Microsystems, Deerfield, IL, USA), and the date was recorded. With a sterile technique, single-spore

colonies $\geq$ 1 mm in diameter were transferred onto a new Petri plate containing the same media (either MEA or MMN) prepared without antibiotics. Several single-spore isolates were taken from the original spore isolation plate to obtain multiple isolates and reduce the likelihood of culturing nontarget species. For details on single-spore isolations, see Karakehian et al. [18]. Briefly, single-spore isolation of fungi requires the location of germinated spores to be marked under a microscope, the spores to be removed under sterile conditions and placed on sterile media.

### 2.3. Culture Preservation and Storage

Each isolate grew for several days before being transferred onto a plate of antibiotic-free media. Mycelium from this plate was used to inoculate a stock slant with the same media. Stock slants were made by sterilizing 35 mL glass vials and filling them with 20 mL of sterilized media cooled at a 45-degree angle. Additional stock cultures were stored using Castellani's method with sterile water [25]. Briefly, 15 mL of sterile deionized water was put into a sterile vial, and four pieces of the fungal colony were cut out of the plate and submerged. Isolates archived by both storage methods were kept at room temperature and in a cold room at 4 °C [26].

### 2.4. Visualization and Microscopy

Ascospores were examined on Petri plate lids and media under 40× or 100× magnification. A study of ascospore morphology at higher magnification was performed by suspending a sterile microscope slide in MEA so that a thin film of media formed on the slide. The slide was removed, and the underside was wiped clean with a Kimwipe. Then, 20 μL of sterile water containing ascospores harvested from Petri plate lids was pipetted onto the media-coated slide and sealed with a sterile cover slip. The slide was then put into a sterile Petri plate containing a Kimwipe soaked in sterile water, and the plate was sealed with Parafilm. Germination was observed the following day.

### 2.5. Molecular Methods

DNA was extracted by suspending a small amount of mycelium in 20 μL of extraction solution (filter-sterilized 1 mL/L of 1 M Tris, 0.186 g/L KCl, 37 mg/L EDTA, set to pH 9.5–10.0 with 1 M NaOH) for 10 min at room temperature (20 °C), followed by cell lysing at 95 °C for 10 min, as previously described [27]. Immediately after lysing, 40 μL of filter-sterilized 3% bovine serum albumin (BSA) was pipetted into each tube. For each PCR amplification reaction, 0.375 μL of 10 μM ITS1F and LR3, 4 μL of ultrapure water, 6.25 μL of DreamTaq™ Green PCR Master Mix (2×) (Thermo Scientific, Waltham, MA, USA), and 1 μL of DNA extract were pipetted into a tube. We chose the ITS1F and LR3 primers because these amplify a region of ribosomal DNA that includes the widely accepted DNA barcode, ITS1-5.8S-ITS2 (ITS), and the first part of the large subunit of the ribosome (LSU), which is useful for phylogenetic placement [28]. PCR reaction conditions were as follows: 35 cycles of denaturing at 95 °C for five minutes and 30 s, annealing at 57 °C for 30 s, and extension at 72 °C for eight minutes, with a final extension at 72 °C for seven minutes.

PCR products were checked for successful amplification using gel electrophoresis. To clean successful amplicons, 3 μL of each PCR product was combined with 2.4 μL of Exo-AP (20 U/μL exonuclease 485 (Sigma, Burlington, MA, USA) and 5 U/μL Arctic phosphatase (New England Biolabs, Ipswich, MA, USA), incubated for 30 min at 37 °C, 15 min at 80 °C, and held at 12 °C until the cleaned PCR products were removed. Cleaned PCR products were then submitted to the Research Technology Support Facility Genomics Core at Michigan State University for Sanger sequencing with the same primers used for PCR amplification.

We manually trimmed each sequence to include only the ITS region and then compared it against the National Center for Biotechnology Information (NCBI) BLASTn database to obtain phylogenetic information on the identity. We also directly compared ITS sequences from cultures with those obtained from direct sequencing of the same fungal specimen. For

molecular-based species delimitation, we assembled an LSU dataset and an ITS dataset for closely related taxa (threshold of 90% identity and 60% coverage) obtained through our BLAST searches. We combined these Pezizales sequences with our sequences generated from fungal cultures and aligned them using MAFFT v 7.471 [29]. Alignments were visualized and examined for accuracy in Se-AL v 2.0a11 [30]. Maximum likelihood analyses were performed for each locus with RAxML-HPC2 v 8.2.12 [31] using the GTRCAT substitution model with 1000 bootstrap replicates. The resulting best tree for each alignment was visualized in FigTree v 1.2.4 (http://tree.bio.ed.ac.uk/software/figtree/, accessed on 1 March 2023). Statistical support for ML was considered significant when bootstrap values were ≥70%. Alignments are available from Open Science Framework using the following link: https://osf.io/sy3kg/files/osfstorage (accessed on 1 March 2023). Culture sequences were deposited in NCBI under the accession numbers: OQ413215–OQ413224, OM672689, OM672856, OM672695, OM672751, OM672687, OM672899, OM672830, OQ413225, and OQ413226 (Table 1).

**Table 1.** Pezizales collections for which spores successfully germinated and were grown in axenic culture. Numbers following taxon names indicate distinct yet unidentified taxa.

| Herbarium Number | Taxon | Collection Date | Date of Inoculation | Days until Germination | GenBank Number |
|---|---|---|---|---|---|
| FLAS-F-68218 | *Galiella rufa* | 16 August 2021 | 14 February 2022 | 5 | OM672689 |
| FLAS-F-68574 | *Galiella rufa* | 12 August 2021 | 17 January 2022 | 7 | OM672856 |
| MSC0286084 | *Gyromitra venenata* | 23 May 2021 | 31 March 2022 | 1 | OQ413215 |
| MSC0286085 | *Morchella angusticeps* | 1 May 2021 | 5 April 2022 | 1 | OQ413216 |
| FLAS-F-68233 | *Peziza griseorosea* | 16 August 2021 | 13 January 2022 | 11 | OM672695 |
| FLAS-F-68375 | *Peziza griseorosea* | 13 August 2021 | 13 January 2022 | 1 | OM672751 |
| MSC0276205 | *Peziza* sp. 2 | 31 March 2021 | 28 March 2022 | 26 | OQ413218 |
| MSC0286082 | *Peziza varia* complex sp. | 23 May 2021 | 12 April 2022 | 2 | OQ413219 |
| MSC0286086 | *Peziza varia* complex sp. | 15 October 2021 | 25 January 2022 | 3 | OQ413220 |
| MSC0286088 | *Peziza varia* complex sp. | 11 October 2021 | 21 February 2022 | 1 | OQ413221 |
| MSC0286087 | *Peziza varia* complex sp. | 17 April 2021 | 28 March 2022 | 1 | OQ413222 |
| MSC0286080 | *Peziza* sp. 1 | 5 October 2020 | 31 March 2022 | 12 | OQ413223 |
| MSC0286083 | *Phylloscypha phyllogena* | 23 May 2021 | 12 April 2022 | 13 | OQ413217 |
| MSC0286081 | *Pseudoplectania* sp. | 23 May 2021 | 28 March 2022 | 1 | OQ413224 |
| FLAS-F-68215 | *Sarcoscypha occidentalis* | 16 August 2021 | 31 January 2022 | 8 | OM672687 |
| FLAS-F-68662 | *Sarcoscypha occidentalis* | 8 September 2021 | 14 February 2022 | 9 | OM672899 |
| FLAS-F-68510 | *Scutellinia* sp. 2 | 7 September 2021 | 31 January 2022 | 15 | OM672830 |
| MSC0286078 | *Scutellinia* sp. 1 | 12 July 2021 | 5 April 2022 | 10 | OQ413225 |
| JV358 | *Sphaerosporella brunnea* | 6 October 2021 | 18 February 2022 | 5 | OQ413226 |

## 3. Results

Out of the 55 collections that were tested in this study, single-spore isolates were generated for 21 representatives (a success rate of 38% overall, and 45% for saprobic collections) from 14 different species including 2 ectomycorrhizal taxa (Table 1). Isolates were made from the following six Pezizales families: Discinaceae, Morchellaceae, Pezizaceae, Pyronemataceae, Sarcosomataceae, and Sarcoscyphaceae (Figure 2).

Some taxa, such as *Morchella angusticeps* and *Pseudoplectania lignicola*, were easily identified because the sequences from the cultures showed ≥99% similarity in the ITS region to reliable reference sequences on GenBank, and the morphology of their ascoma was congruent with definitive morphological characteristics [2,32]. For less straightforward identifications, we created seven separate nucleotide alignments for further phylogenetic analysis. These separate alignments were made to phylogenetically place isolates [2]. Isolates identified in this manner were determined to be *Galiella rufa*, *Gyromitra venenata*, *Peziza griseorosea*, two unnamed species of *Peziza sensu stricto* (designated here as *Peziza* sp. 1 and sp. 2), *Phylloscypha phyllogena*, *Sarcoscypha occidentalis*, two unnamed species of *Scutellinia sensu stricto* (designated here as *Scutellinia* sp. 1 and sp. 2), and *Sphaerosporella*

*brunnea.* In total, phylogenetic analyses allowed us to identify 14 species belonging to nine genera and obtained as isolates from ascospore harvesting and germination (Table 1; Supplementary Information, Figures S1–S8).

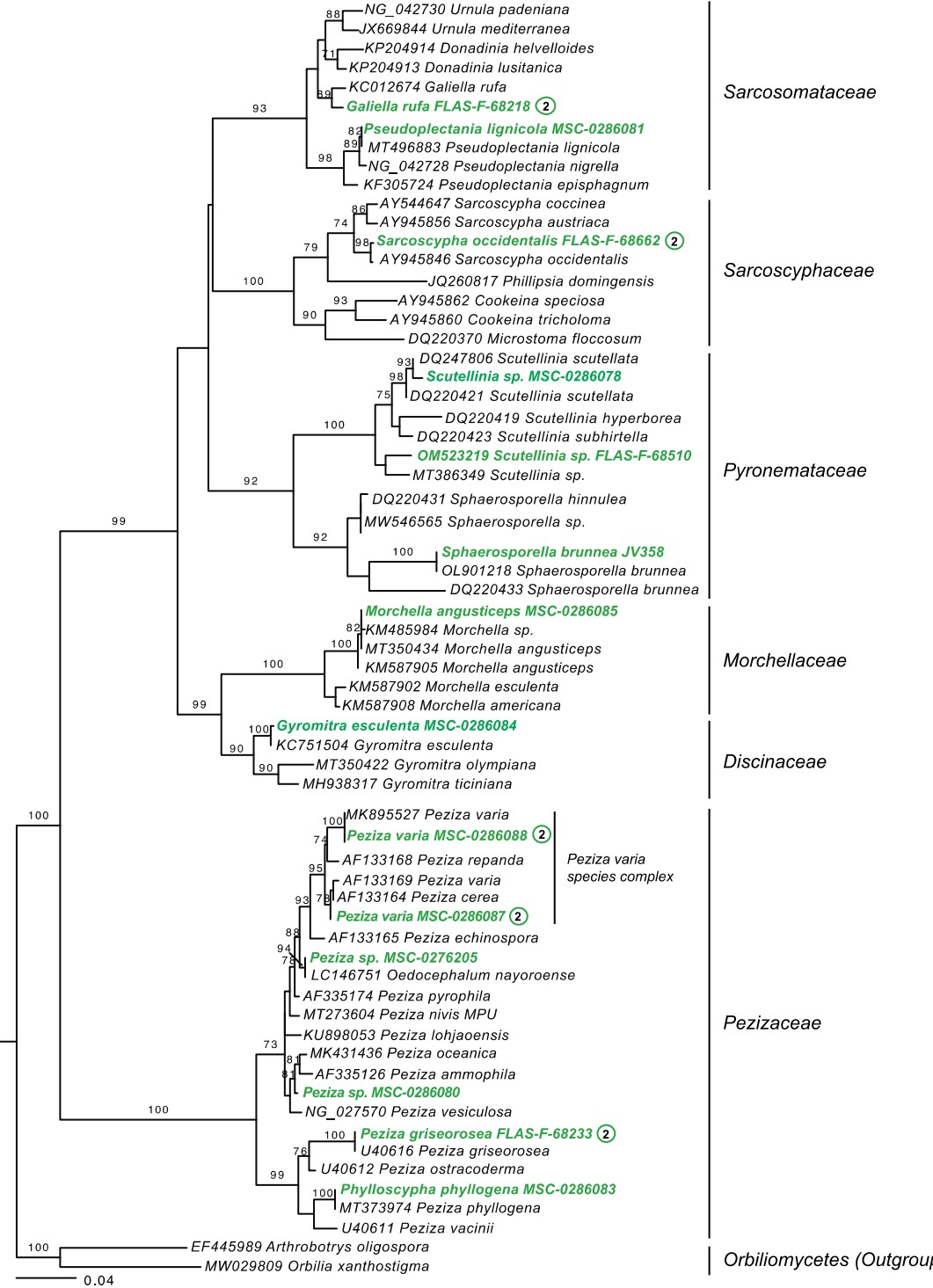

**Figure 2.** One of the most likely RAxML phylogenetic trees estimated from an LSU alignment showing the wide diversity of single-spore isolates among other Pezizales fungi. Significant support is denoted by bootstrap values ≥70% at nodes. Green taxa represent isolates that germinated and grew in axenic cultures in this study. Circled numbers represent the number of different isolates obtained of a given species.

Most of the isolates generated from ascospores belonged to saprotrophic taxa. However, two ectomycorrhizal species, *Sphaerosporella brunnea* and *Peziza griseorosea*, germinated and were successfully grown in culture. An additional 11 ectomycorrhizal species did not germinate on MMN media under laboratory conditions (Supplementary Information Table S1). In contrast, 12 of the 19 tested saprotrophic species (63%) successfully germinated on both WA and MEA (Table S1).

With this approach, we also obtained new information about the process of spore germination and the formation of other spore types from axenic cultures. Germination from ascospores in most Pezizales is more typically initiated by one or two germ tubes (Paden 1972). However, *Gyromitra* ascospores produced up to eight germ tubes when germinating at low spore density, but only produced one or two when germinating near other spores (Figure 3). Furthermore, mitospores formed in cultures of species within the *Peziza varia* complex, arthroconidia were formed by *Phylloscypha phyllogena*, and chlamydospores were produced by *Morchella angusticeps* and *Scutellinia* sp. 1 and 2 (Figure 4). To the best of our knowledge, this is the first report of chlamydospores in cultures of *Morchella angusticeps* (Figures S9–S11).

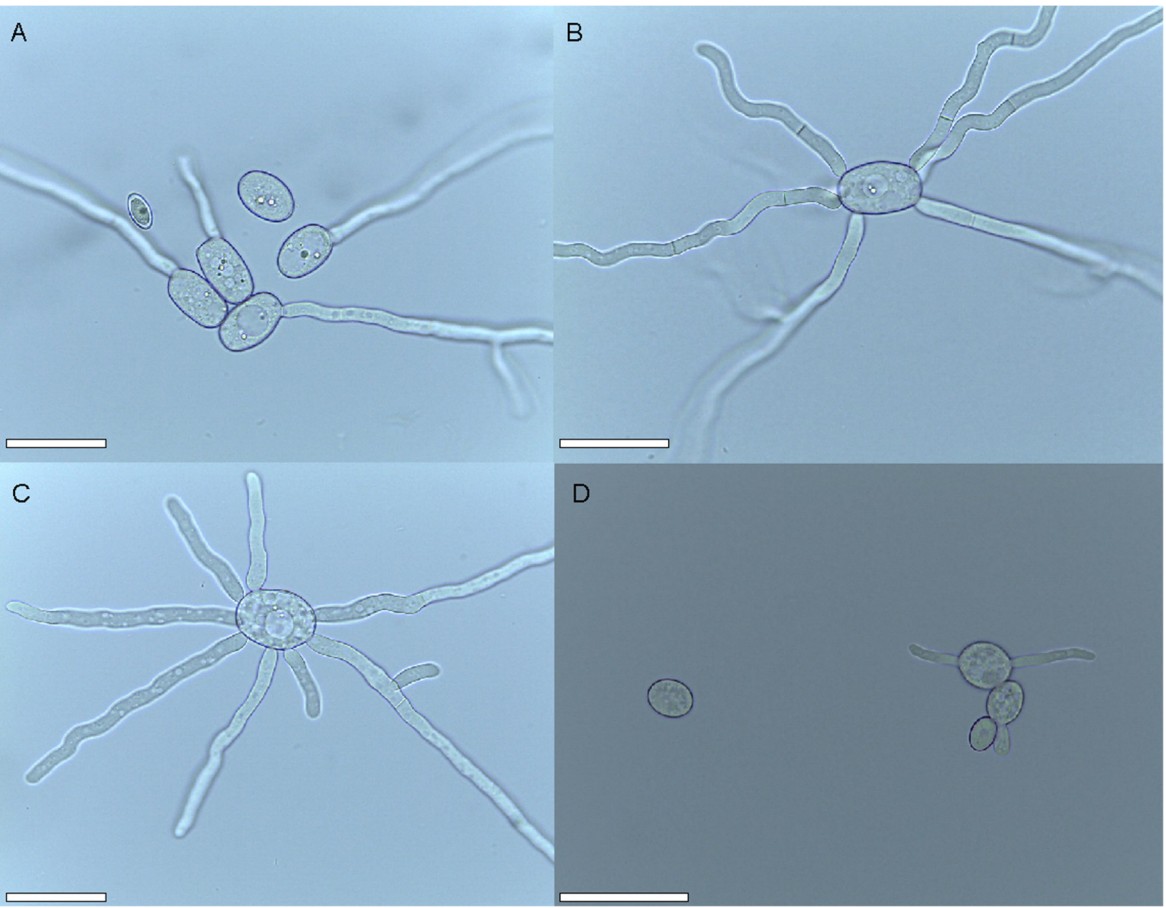

**Figure 3.** *Gyromitra venenata* (MSC0286084) ascospores at 400× magnification. (**A–C**) Ascospores germinating, exhibiting dramatic swelling and multipolar germ tube formation. (**D**) Dormant and germinating ascospores. Bar = 50 μm.

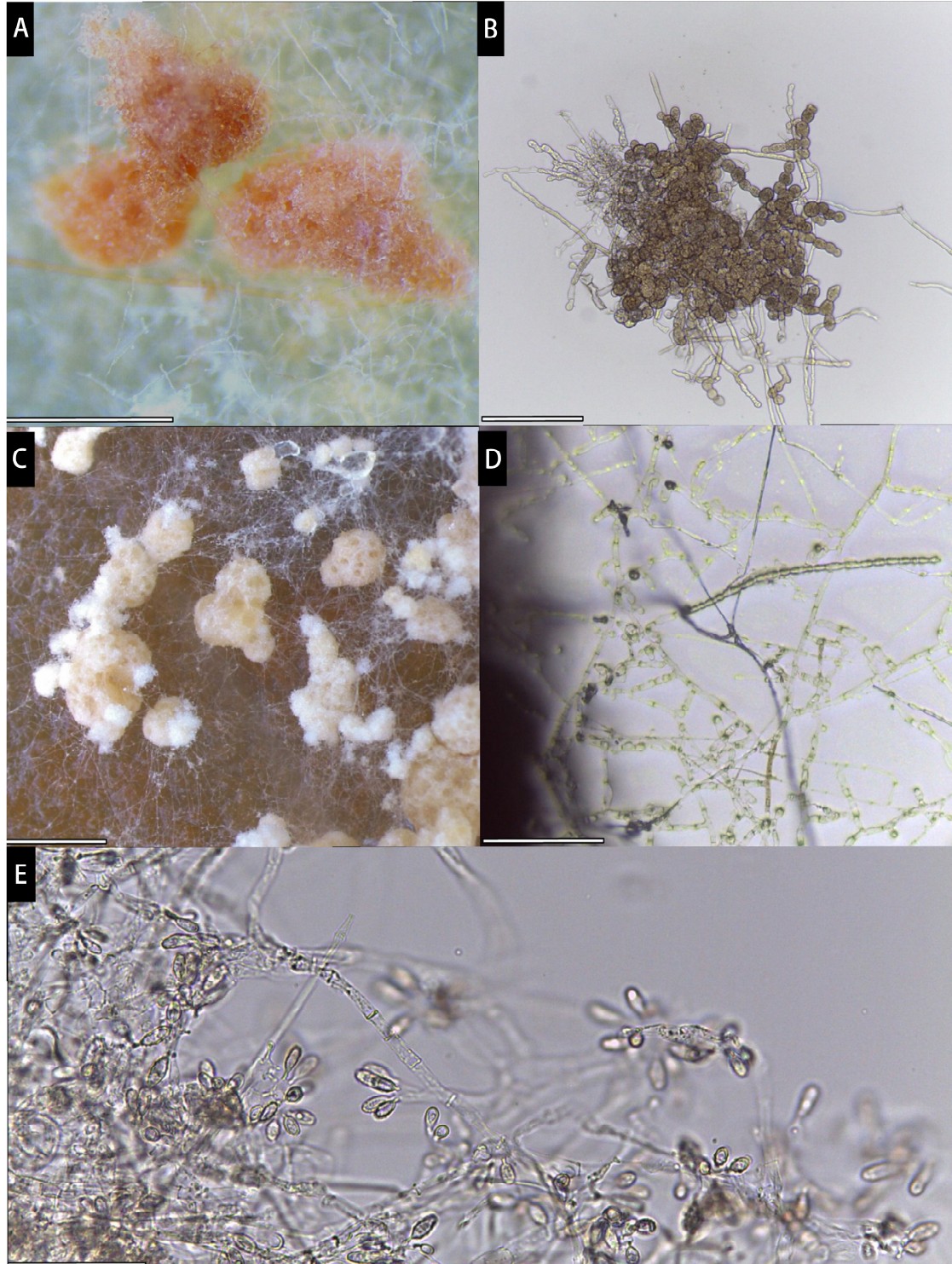

**Figure 4.** Spores and other survival structures that formed in Pezizales cultures. (**A**) Microsclerotia in *Morchella angusticeps* (MSC0286085). (**B**) Chlamydospores of *Morchella angusticeps* (MSC0286085). (**C**) Sclerotia of *Phylloscypha phyllogena* (MSC0286083), which only formed in multispore cultures. (**D**) Arthroconidia of *Phylloscypha phyllogena* (MSC0286083). (**E**) Conidia in *Peziza* sp. 1 (MSC0286080). Scale bars: (**A**) = 790 µm; (**B**) = 200 µm; (**C**) = 3.175 mm; (**D**) = 200 µm; (**E**) = 50 µm.

## 4. Discussion

Here, we described a method for collecting Pezizales ascospores in the field for later germination and growth. We demonstrated its effectiveness by obtaining single-spore

isolates from apothecial Pezizales collections of several saprotrophic and ectomycorrhizal species. This technique is particularly useful when collections are made in locations where prompt culturing is not possible. Prepared with sterile Petri plates, scientists or citizen scientists can quickly set up ascomata for spore capture during collecting in the field. The resulting dried ascospores on the lid can be preserved for culturing in a controlled laboratory environment by scientists weeks, months, or even years later. This technique has the added benefit of keeping collections intact, preserving the quality and quantity of herbarium collections.

Following this method, researchers may obtain single-spore isolates from diverse species with more control and precision than is possible by collecting ascospores directly onto axenic media for germination and growth. This approach can be used to add further value to new fungarium collections and, if provided with sterile Petri plates, can easily be incorporated into efforts of citizen scientists and students to assist with fungal biology research and collections by allowing spores to be obtained to accompany useful collections for science. Field capture of ascospores is especially desirable when collecting undescribed or rare taxa. This field method is especially well suited for saprotrophic Pezizales and has promise to help facilitate future studies of Pezizales fungi and their biology, including mycelial development, symbiotic interactions in culture, and genomic and transcriptomic data.

We note that the 35 collections for which we were unable to obtain isolates included a mycoparasite, a parasite of bryophytes, 15 putative saprotrophs, and 18 ectomycorrhizal fungi from the families Helvellaceae, Otideaceae, Pulvinulaceae, and Wynneaceae—which are rarely isolated. These families consist primarily of obligately biotrophic fungi or fungi with an unknown trophic mode. Obligate biotrophs and ectomycorrhizal taxa are notoriously difficult to isolate from ascospores, often requiring the presence of a host for growth factors because of various types of auxotrophy, or specialized media (e.g., [33–36]), or heat [8], or cold treatments [37]. Knowledge gleaned from ectomycorrhizal basidiospore germination studies, such as the use of *Rhodotorula glutinis* to induce germination [38], or reduction in media-inhibiting factors, like ammonium [39] or organic acids produced during the autoclaving of agar [40], may increase the success rate of ectomycorrhizal ascospore germination in the future. Harvested ascospores can be tested on various types of media, at different temperatures, and in the presence of different organisms. Although saprotrophic taxa had a higher success rate than ectomycorrhizal taxa, we were still unable to culture many collections. It is well known that some saprotrophs require environmental cues or host factors that were not present in our general media [41]. Regardless of trophic state, spores may not be viable for a variety of reasons that we did not investigate. In such cases, vital stains could be used to assess spore viability before attempting germination [41–43].

To the best of our knowledge, this is the first time that *Peziza griseorosea* has been cultured [44]. *Peziza griseorosea* is not closely related to the saprobic lineage *Peziza* sensu stricto, and new nomenclatural changes are needed to better reflect the phylogenetic relationships in this group [2]. The trophic habit of *Peziza griseorosea* has not been previously characterized, but three ectomycorrhizal root tip sequences (two from *Quercus* and one from *Fagus*) indicate that this species is ectomycorrhizal (Figure S3). In addition, sister lineages *Galactinia* and *Legaliana* are ectomycorrhizal, which further supports the hypothesized ectomycorrhizal habit of *P. griseorosea* [2,45]. We used this single-spore isolate to generate high-quality genomic data for this species.

One of the lines of inquiry that can be investigated with cultures of Pezizales is the documentation of spore germination and differentiation of hyphae in anamorphs and other structures in culture. For example, the germination of most Pezizales ascospores commences with the formation of one to three (e.g., [46–48]) or four germ tubes (e.g., [49]. Thus, our documentation of eight germ tubes produced by *Gyromitra venenata* in low-density cultures is remarkable (Figure 3). Boedijn [50] illustrated many conidiophores rising from ascospores of *Cookeina sulcipes*, but none of them appeared to elongate into hyphae. We speculate that the low density of spores may trigger the release of multiple

germ tubes for the rapid colonization of the substrate. However, we do not know how common this is in the Pezizales, or if it occurs in nature. Chlamydospores were observed in our *Morchella angusticeps* cultures, and these structures were also reported by Alvarado-Castillo et al. [51] in *Morchella esculenta* and *M. conica* cultures and by Yuan et al. [52] in *M. sextelata* cultures. Yuan et al. [52] hypothesized that these structures may be important for *Morchella* species to survive suboptimal conditions. Thus, they may be an important aspect of the *Morchella* lifecycle and deserve further scrutiny.

When collecting fungal ascospores, it is important to consider that immature and overmature ascomata may release very few or no ascospores [18]. Inconsistencies in spore release can be ameliorated by collecting ascomata in mature and healthy conditions or inducing pressure changes by repeatedly opening and closing the lid of the Petri plate, cooling the environment, or leaving samples in Petri plates for at least an hour. However, ascomata left for too long in a plate may release an overabundance of spores and may cause the plates to become too humid. Also, if not properly dried, spores may prematurely germinate and thus reduce the number of viable spores for later use. Furthermore, dense spore prints require further serial dilution to procure single-spore isolates. Plating spores in a high density appeared to inhibit spore germination and affect germ tube growth in some species, as also reported by Karakehian et al. [18]. This was clearly observed with *Gyromitra* ascospores, which produced fewer germ tubes when in close proximity to other spores. Inhibition of spore germination with high spore density may be a mechanism of ensuring spores have resources to germinate or may simply be the effect of nutrient availability on each spore (Figure 3) [53]. High spore density can be avoided by removing the ascomata within 24 h of collection or diluting the spore slurry.

Another consideration when using the method described here relates to the presence of nontarget spores in some spore prints. We unintentionally cultured hymenium-colonizing mycoparasites during the course of this study, including species of *Acremonium* [54] and unidentified Sordariomycetes that were present on the substrate (see Figure 1A). The presence of nontarget spores can be limited by removing as much soil and plant debris as possible before placing the ascoma of interest in the Petri plate when performing collections. In some cases, especially with dung-associated Pezizales, ascomata are small and cannot be removed from the substrate without destroying the collection. In such cases, special care should be taken to focus spore dispersal as described above, limit movement of the sample, and limit the amount of time that the ascomata are in a plate.

Additionally, we found that plating spores on more than one media type was often redundant and unnecessary. We agree with Karakehian et al. [18] that using water agar or other low-nutrient media with antibiotics is optimal. Low-nutrient agar slows the growth of mycelium and gives the researcher more time to isolate colonies from single spores before individuals grow over one another.

Finally, while we focused solely on culturing and collecting Pezizales, this procedure can be employed for most fungi that form asci and actively discharge their spores. For instance, we also applied this method successfully for harvesting ascospores from species of Leotiomycetes, Sordariomycetes, and Lecanoromycetes. Thus, with this approach, species of undescribed and poorly documented ascomycetes can be collected, cultured, and studied.

In conclusion, we presented a simple and effective method for harvesting ascospores from fresh Pezizales collections in the field, thereby allowing spores to be cultured from collections at a later date. This method is accessible to citizen scientists and others, providing an additional avenue for contributing to fungal biology research in addition to collecting ascomata and photographs. However, researchers would need to provide collection materials (sterile Petri plates, tape, some simple instructions, and perhaps a pre-paid padded envelope addressed to the research laboratory) to collaborators for making spore collections. With this method, we were able to obtain single-spore isolates from six families of Pezizales, including species known to be ectomycorrhizal and clades that had not been previously grown in axenic culture. Following this approach, researchers may be able

to obtain isolates of previously uncultured fungi, allowing studies in development and growth of these fungal species to address questions related to their biology, evolution, and ecogenomics.

**Supplementary Materials:** The following supporting information can be downloaded at: https://www.mdpi.com/article/10.3390/d16030165/s1, Figure S1: Most likely RAxML phylogenetic tree showing placement of single-spore isolates among other *Galiella* isolates. Significant support is denoted by bootstrap values ≥70% at nodes. Green taxa represent isolates that germinated and grew in axenic culture for this study; Figure S2: Most likely RAxML phylogenetic tree showing placement of single-spore isolates among other *Peziza* isolates. Significant support is denoted by bootstrap values ≥70% at nodes. Green taxa represent isolates that germinated and grew in axenic culture for this study; Figure S3: Most likely RAxML phylogenetic tree showing placement of single-spore isolates among other *Peziza griseorosea* isolates. Significant support is denoted by bootstrap values ≥70% at nodes. Green taxa represent isolates that germinated and grew in axenic culture for this study; Figure S4: Most likely RAxML phylogenetic tree showing placement of single-spore isolates among other *Phylloscypha* isolates. Significant support is denoted by bootstrap values ≥70% at nodes. Green taxa represent isolates that germinated and grew in axenic culture for this study. Figure S5: Most likely RAxML phylogenetic tree showing placement of single-spore isolates among other *Sarcoscypha* isolates. Significant support is denoted by bootstrap values ≥70% at nodes. Green taxa represent isolates that germinated and grew in axenic culture for this study; Figure S6: Most likely RAxML phylogenetic tree showing placement of single-spore isolates among other *Scutellinia* isolates. Significant support is denoted by bootstrap values ≥70% at nodes. Green taxa represent isolates that germinated and grew in axenic culture for this study; Figure S7: Most likely RAxML phylogenetic tree showing placement of single-spore isolates among other *Sphaerosporella* isolates. Significant support is denoted by bootstrap values ≥70% at nodes. Green taxa represent isolates that germinated and grew in axenic culture for this study; Figure S8: Most likely RAxML phylogenetic tree showing placement of single-spore isolates among other *Gyromitra* isolates. Significant support is denoted by bootstrap values ≥70% at nodes. Green taxa represent isolates that germinated and grew in axenic culture for this study; Figure S9: *Phylloscypha phyllogena* (MSC0286083) sclerotia and chlamydospores forming on MEA. The images are from the intiial plate and spores were able to interact. (A). Sclerotia. (B). Sclerotia. (C). Chlamydospores Bar = 20 μm. (D). Chlamydospores. Scale Bars: (A) = 1.5 mm, (B) = 800 μm, (C) = 20 μm, (D) = 5 μm; Figure S10: *Morchella angusticeps* (MSC0286085) chlamydospores forming from a single spore isolate on MEA. (A). Sclerotia. (B). Chlamydospores. (C). Chlamydospores Bar = 50 μm. (D). Chlamydospores. Scale Bars: (A) = 1.6 mm (B) = 200 μm (C,D) = 50 μm; Figure S11: Peziza sp. 1 (MSC0286080) conidia that formed from a single spore isolate growing on MMN. Scaler Bars: (A–C) = 50 μm; Table S1: All Pezizales fungi collected and tested for germination and growth in axenic culture. Taxa for which no germination date is listed did not successfully germinate in culture.

**Author Contributions:** Conceptualization, A.S., J.V.W., B.L., R.H., M.E.S. and G.B.; Methodology, A.S., J.V.W., B.L., R.H., M.E.S. and G.B.; data curation, A.S., J.V.W. and R.H.; formal analysis, A.S. and J.V.W.; writing-original draft preparation A.S. and J.V.W.; writing-review and editing, A.S., J.V.W., B.L., R.H., M.E.S. and G.B.; visualization, A.S.; supervision, J.V.W.; project administration G.B., M.S. and R.H.; funding acquisition, G.B., M.S. and R.H. All authors have read and agreed to the published version of the manuscript.

**Funding:** This project was supported by a U.S. National Science Foundation (NSF) grant DEB-1946445 (to M.E.S., R.H., and G.B.), an NSF Research Traineeship Program (DGE-1828149 to J.V.), NSF REU support to A.S. (DEB 1737898), and an NSF Graduate Research Fellowship to B.L. (no. 2019277707).

**Institutional Review Board Statement:** Not applicable.

**Data Availability Statement:** Data are contained within this article and Supplementary Materials.

**Acknowledgments:** This project was part of a joint undergraduate student/graduate student project initiated as a research experience for undergraduates (REU) study by A.S. and J.V.W. at Michigan State University as part of NSF grant DEB-1946445. Collecting in the Great Smoky Mountain National Park was conducted under permit no. GRSM-2021-Sci-2516; collections in Nantahala National Forest were made with written permission from the United States Forest Service. We thank Paul Super, Rachel Sweeney, and Brandon Matheny for guidance regarding collection locations in the Great Smoky Mountains National Park and Maria Dunlavey for guidance on collecting in the Nantahala

National Forest. We thank Alden Dirks and Bryan Rennick for their collections used in this study. We thank the staff at MSU RTSF for generating sequence data and both the University of Florida Fungal Herbarium (FLAS-F) and Michigan State University Herbarium (MSC) for accessioning collections and making them available for study.

**Conflicts of Interest:** The authors declare no conflicts of interest.

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
