# Peer review of "Effective Field Collection of Pezizales Ascospores for Procuring Diverse Fungal Isolates"

_diversity, doi:10.3390/d16030165_

Round 1
Reviewer 1 Report
Comments and Suggestions for Authors
The study is interesting and potentially useful for the scientific community. Interestingly, the Authors refer to the "citizen scientists" approach, which is emerging as an important branch of future ecological studies.
I have only a few minor questions.
1.
In lines 262-264, the Authors mentioned Table S1, but this table is absent in the manuscript files, so the list of 15 analysed EM fungi is missing as well. Please, upload this table.
2.
In the lines 23-25, the Authors wrote:
"This method worked well for 23 saprotrophic taxa (12 out of 19 species, 63%), and was effective for a few ectomycorrhizal taxa (2 out of 13 species, 15%)."
This is an interesting result.
What is the reason for the low efficiency of the method for EM fungi? Could you discuss it against the background of other studies on spore germination for EM fungi? Preferably Ascomycota, but also Basidiomycota, if necessary.
3.
The discussion section should be improved. The Authors involved a lot of background data (current state of knowledge), which should rather be replaced in the Introduction section, but other studies on the germination of fungal sporocarps are barely mentioned and discussed.
Here are a few additional questions, that would help:
- how many different methods for fungal spore germination (for all fungi, and specifically for EM fungi) are used (or published) and why did you select/develop your method?
- how does the efficiency of spore germination (63% and 15% for saprotrophic and EM fungi) in this study stack up against the efficiency provided by other studies?
- what is the novelty of this study, compared to the previous studies on fungal sporocarp germination?
- which factors can be responsible for the unsuccessful germination? Are you able to improve/eliminate those factors, and improve the method in this way?
Author Response
Thank you for your careful review of our manuscript. We appreciated the reviews and suggestions and have addressed these comments in the enclosed revisions and point-by-point below. We feel that they strengthen this manuscript. We hope you find our responses satisfactory and our revised manuscript acceptable for publication in Diversity.
REVIEWER 1
The study is interesting and potentially useful for the scientific community. Interestingly, the Authors refer to the "citizen scientists" approach, which is emerging as an important branch of future ecological studies.
I have only a few minor questions.
In lines 262-264, the Authors mentioned Table S1, but this table is absent in the manuscript files, so the list of 15 analysed EM fungi is missing as well. Please, upload this table.
Response: Table S1 is now included in the manuscript submission. Thank you for catching this error.
2. In the lines 23-25, the Authors wrote:
"This method worked well for 23 saprotrophic taxa (12 out of 19 species, 63%), and was effective for a few ectomycorrhizal taxa (2 out of 13 species, 15%)."
This is an interesting result. What is the reason for the low efficiency of the method for EM fungi? Could you discuss it against the background of other studies on spore germination for EM fungi? Preferably Ascomycota, but also Basidiomycota, if necessary.
Response: We discuss this in lines 322-326. “Obligate biotrophs and ectomycorrhizal taxa are notoriously difficult to isolate from ascospores, often requiring the presence of a host for growth factors because of various types of auxotrophy, or specialized media (e.g. Iotti et al. 2013; Domínguez Romero et al. 2013; Luginbuehl et al. 2017; Lorrain et al. 2019), or heat (Paden 1972), or cold treatments (Ogawa et al. 2000). Harvested ascospores can be tested on various types of media.”. This paper describes a general method that can be simply applied to media we did not test.
These papers bring up how adjustments to media can improve efficiency of some specific organisms. 326-332. “Knowledge gleaned from ectomycorrhizal basidiospore germination studies, like the use of Rhodotorula glutinis to induce germination (Fries 1943), or reduction of media inhibiting factors, like ammonium (Fries 1976) or organic acids produced during the autoclaving of agar (Fries 1978), may increase the success rate of ectomycorrhizal ascospore germination in the future. Harvested ascospores can be tested on various types of media, at different temperatures, and in the presence of different organisms.”
The discussion section should be improved. The Authors involved a lot of background data (current state of knowledge), which should rather be replaced in the Introduction section, but other studies on the germination of fungal sporocarps are barely mentioned and discussed.
Here are a few additional questions, that would help:
- how many different methods for fungal spore germination (for all fungi, and specifically for EM fungi) are used (or published) and why did you select/develop your method?
- how does the efficiency of spore germination (63% and 15% for saprotrophic and EM fungi) in this study stack up against the efficiency provided by other studies?
- what is the novelty of this study, compared to the previous studies on fungal sporocarp germination?
- which factors can be responsible for the unsuccessful germination? Are you able to improve/eliminate those factors, and improve the method in this way?
Response: Thank you for these suggestions. We have restructured the discussion to take these points into consideration. Many factors influence spore germination success rates and we speculate why some saprotrophic and ectomycorrhizal taxa were unsuccessful in lines 332-338. “Although saprotrophic taxa had a higher success rate than ectomycorrhizal taxa, we were still unable to culture many collections. It is well known that some saprotrophs require environmental cues or host factors that were not present in our general media (Miller et al. 1993). Irregardless of trophic state, spores may not be viable for a variety of reasons that we did not investigate, we propose using vital stains to assess viability before attempting germination (Miller et al. 1993, Ishida et al. 2008, Bohorquez et al. 2021)."
Our method is novel because it allows for the capture of spores while in the field, and the storage of ascospores from many species whereas previous approaches release spores onto media. We now say on lines 74-79 “Volk and Leonard (1990) mentioned placing bisected Morchella ascomata in Petri dishes to collect spores en masse for generating cultures. Karakehian et al (2021) presented step-by-step methods for culturing ascomycete fungi directly from ascospores by placing a section of a mature ascoma in an inverted Petri plate of nutrient media, and allowing the ascospores to be ejected directly onto suitable growth media. While this method is useful for generating pure cultures, it still requires media and axenic conditions to be available when collections are fresh.”
We discuss the factors that could be responsible for unsuccessful germination here. 322-326 Obligate biotrophs and ectomycorrhizal taxa are notoriously difficult to isolate from ascospores, often requiring the presence of a host for growth factors because of various types of auxotrophy, or specialized media (e.g. Iotti et al. 2013; Domínguez Romero et al. 2013; Luginbuehl et al. 2017; Lorrain et al. 2019), or heat (Paden 1972), or cold treatments (Ogawa et al. 2000).
Some of the factors above suggest methods to induce spore germination, such as by using host growth factors and temperature treatments. We suggest here that some methods found to work for germinating some more difficult ECM basidiomycete spores may work for ECM ascospores as well. 326-332. Knowledge gleaned from ectomycorrhizal basidiospore germination studies, like the use of Rhodotorula glutinis to induce germination (Fries 1943), or reduction of media inhibiting factors, like ammonium (Fries 1976) or organic acids produced during the autoclaving of agar (Fries 1978), may increase the success rate of ectomycorrhizal ascospore germination in the future. Harvested ascospores can be tested on various types of media, at different temperatures, and in the presence of different organisms.

Reviewer 2 Report
Comments and Suggestions for Authors
Dear authors and editors,
The article "Effective filed collection of Pezizales ascospores for procuring diverse fungal isolates" is a very interesting research paper detailing the in vivo method of collecting fungal spores of the order Pezizales, and definitely is recommended for publication after a minor revision.
Below are few comments which I hope will help the authors to improve the manuscript and convey their main message.
In general
A couple of times, the authors have stated that the techniques they described, can be used by citizen scientists. Nevertheless, the procedure is quite complex for a non-scientist and involves special preparation (such as Petri dishes and culture media availability, identification of species of interest in the field, etc.). Could authors explain in more detail how they plan implement the data from the citizen scientists? Also, in Table 1, I see that some individuals were only identified to the genus level. Could the authors explain why? This fact immediately raises the question: "What is the scientific merit of collecting spores and cultivating/observing fungal cultures without knowing the actual species?" What could be a solution for this?
In more detail
Figure 1.
It is not completely clear, what one should see at the Figure 1E & F. Please include more descriptive legend.
Lines 140-143
Could you explain the choice of media in more detail? Why did you use different media for different groups of fungi?
Line 147-177
It is not clear from the text whether the procedures were carried out under sterile conditions? For example, in lines 150-151, observations were made under a compound microscope at 100x magnification to ensure that the target spores were suspended in water. Did you open the lids of the Petri dishes when examining the spores? Then: "In these cases, the spores were first disturbed with a pipette tip, withdrawn into the pipette tip, and then added to another 100 μl of sterile deionized water". Did you place them into the Eppendorf tube? Into another Petri dish? etc... Please make sure to be as precise as possible in the method description to ensure the reproducibility of your methods.
Lines 176-177
Please include a brief description of the Karakehian et al. 2021 technique, as you have done for other methods. This will make it easier to read the following paragraph.
Line 188 “Ascospores were visualized”
You mean examined? Or did you take pictures of all isolates?
Line 215 – “We manually trimmed each sequence to include only the ITS region”
What did you use as a reference? How did you identify the beginning of the ITS? Did you use the entire ITS or just ITS1?
Lines 216-217 – “…then compared it against the National Center for Biotechnology Information (NCBI) BLASTn data base to ensure that each culture was from the ascoma of interest.”
Does this mean that all target "taxa" were present on NCBI? What was the percentage of similarity to the best match that you considered significant? Or were all sequences with 100% blast hits?
Line 218 – “we assembled an LSU dataset and an ITS dataset for similar taxa”
What do you mean by "similar taxa"? Perhaps sequences with the best/closest hits?
Line 219 – “We assembled these Pezizales sequences with our sequences generated from fungal cultures”
Assembled or combined? An assembly represents an alignment of one individual/representative.
Line 221 – “Alignments were manually improved in Se-AL v 2.0a11 (Rambaut 2007).”
What do you mean by "improvement"? Did you trim a part of the alignment that could not be aligned, or something else?
Line 222
Could you please explain why the phylogenetic tree is necessary for your work? Wouldn't it be sufficient to do a simple comparison with the BLAST database to make sure that your sequence is close to the ascoma you are interested in? Why did you choose the LSU, and not ITS tree? Were the ITS and LSU phylogenies congruent?
Line 279, Figure 4
It seems the figure is not correctly oriented. Please correct.
Lines 293-301
As mentioned at the beginning, the process described in the manuscript is quite complex and requires special preparation. Please either provide more details about how you plan to use the citizen scientists' data (e.g. additional photos, location data, etc.), or put less emphasis on the citizen science and more on the scientific benefit of your research.
Author Response
Dear authors and editors,
The article "Effective field collection of Pezizales ascospores for procuring diverse fungal isolates" is a very interesting research paper detailing the in vivo method of collecting fungal spores of the order Pezizales, and definitely is recommended for publication after a minor revision.
Below are few comments which I hope will help the authors to improve the manuscript and convey their main message.
Thank you for your careful review of our manuscript. We appreciated the reviews and suggestions and have addressed these comments in the enclosed revisions and point-by-point below. We feel that they strengthen this manuscript. We hope you find our responses satisfactory and our revised manuscript acceptable for publication in Diversity.
- In general, a couple of times, the authors have stated that the techniques they described, can be used by citizen scientists. Nevertheless, the procedure is quite complex for a non-scientist and involves special preparation (such as Petri dishes and culture media availability, identification of species of interest in the field, etc.). Could authors explain in more detail how they plan implement the data from the citizen scientists? Also, in Table 1, I see that some individuals were only identified to the genus level. Could the authors explain why? This fact immediately raises the question: "What is the scientific merit of collecting spores and cultivating/observing fungal cultures without knowing the actual species?" What could be a solution for this?
Response:
In lines 409-410 we clarify that citizen scientists are meant to obtain collections and researchers carry out the culturing and molecular methods.”Researchers can provide collection materials (sterile Petri plates, tape, some simple instructions, and perhaps a pre paid, padded envelope addressed to the research lab) to citizen scientists who can then make spore collections”
This paper was written in part because there is a need to clarify the phylogeny of genera in Pezizales. Having cultures allows us to generate genetic data that can help with this work. Specifically, Peziza is a large polyphyletic genus with many species that are not described or species limits not understood. Consequently we were not able to identify potentially novel Peziza sp collections to the species level. The same is true of Scutellinia.
In more detail
- Figure 1. It is not completely clear, what one should see at the Figure 1E & F. Please include more descriptive legend.
Response: Figure 1 E and F are meant to show the reader what spores look like on petri dish lids after collection. We also show how using markers is an efficient way to show where spores are when microscopes are unavailable. We clarify this in lines 140-143. “E. Confirming presence of Legliana sp. (FLAS-F-68766) spores within the marked circle shown in C; note that spores are present inside the circled area but are not present outside of the border. F. Confirming presence of Scutellinia sp. (FLAS-F-68459) spores within the marked circle as shown in D. ”
- Lines 140-143 - Could you explain the choice of media in more detail? Why did you use different media for different groups of fungi?
Response: We use different media because ECM fungi will not germinate on general media and require specific nutrients. We clarify this on lines 149-152 by saying “Spores of all specimens were also plated on 10% water agar (WA). Saprotrophic fungi are known to generally grow well on MEA whereas ectomycorrhizal fungi often need specific media and supplemented vitamins to grow well, such as MMN (Marx 1969). ”
- Line 147-177 - It is not clear from the text whether the procedures were carried out under sterile conditions? For example, in lines 150-151, observations were made under a compound microscope at 100x magnification to ensure that the target spores were suspended in water. Did you open the lids of the Petri dishes when examining the spores?
Response: We now clarify that these steps are done before working in sterile conditions on lines 157-163. We say “Before plating in sterile conditions, spore prints were visualized under 40X magnification and the area with the highest density of spores was circled with a permanent marker. Then 100 µL of sterile deionized water was gently pipetted up and down multiple times within the marked area. Because spores are sometimes not visible to the naked eye, observations of 10 µl of each spore slurry was deposited on a slide and viewed with a compound microscope at 100X magnification to ensure target spores were suspended in the water.”
- Then: "In these cases, the spores were first disturbed with a pipette tip, withdrawn into the pipette tip, and then added to another 100 μl of sterile deionized water". Did you place them into the Eppendorf tube? Into another Petri dish? etc... Please make sure to be as precise as possible in the method description to ensure the reproducibility of your methods.
Response: We now clarify that water is placed into another tube by saying “Ascospores can clump together in water, making it difficult to create a spore suspension (Hunter et al. 1982). In such cases, spores were first disturbed with a pipette tip, retracted into the pipette tip, and then placed into another 100 µL of sterile deionized water inside of a sterile 1.5 mL tube.”.
- Lines 176-177 - Please include a brief description of the Karakehian et al. 2021 technique, as you have done for other methods. This will make it easier to read the following paragraph.
Response: Thank you for the suggestion, we now briefly describe the Karakehian et al. 2021 technique in lines 188-191 “For details on single-spore isolations see Karakehian et al. 2021. Briefly, single-spore isolation of fungi requires the location of germinated spores to be marked under a microscope, the spores to be removed under sterile conditions, and placed on sterile media. ”
- Line 188 “Ascospores were visualized” - You mean examined? Or did you take pictures of all isolates?
Response: Thank you for this suggestion. We change visualized to examined; “Ascospores were examined on Petri plate lids and media using a Lecia DM750 compound microscope at 40X or 100X magnification. ”
- Line 215 – “We manually trimmed each sequence to include only the ITS region” What did you use as a reference? How did you identify the beginning of the ITS? Did you use the entire ITS or just ITS1?
Response: Thank you for the questions, we identify the ITS region by comparing sequences to known primer sequences. We also clarify why we chose this region and what portion of the sequence we use in lines 219-222 “We chose the ITS1F and LR3 primers because these amplify a region of ribosomal DNA that includes the widely accepted DNA barcode, ITS1-5.8S-ITS2 (ITS), and first part of the large subunit of the ribosome (LSU) which is useful for broad range phylogenetic placements (Schoch et al. 2012).”
- Lines 216-217 – “…then compared it against the National Center for Biotechnology Information (NCBI) BLASTn data base to ensure that each culture was from the ascoma of interest.” Does this mean that all target "taxa" were present on NCBI? What was the percentage of similarity to the best match that you considered significant? Or were all sequences with 100% blast hits?
Response: All target taxa were present on NCBI because original ascoma collections were sequenced and accessioned on the database. Significant best match similarity was considered significant at 99% similarity, if there were no significant matches, we constructed phylogenetic trees. This has been clarified in the text, lines 234-237 say “compared it against the National Center for Biotechnology Information (NCBI) BLASTn database to obtain phylogenetic information on the identity. We also directly compared ITS sequences from cultures with those obtained from direct sequencing of the same fungal specimen. For molecular-based species delimitation, we assembled an LSU dataset and an ITS dataset for closely related taxa (threshold of 90% identity and 60% coverage) obtained through our BLAST searches.”
- Line 218 – “we assembled an LSU dataset and an ITS dataset for similar taxa”. What do you mean by "similar taxa"? Perhaps sequences with the best/closest hits?
Response:We clarify that this means closely related taxa in lines 237-239, “For molecular-based species delimitation, we assembled an LSU dataset and an ITS dataset for closely related taxa (threshold of 90% identity and 60% coverage) obtained through our BLAST searches. ”
- Line 219 – “We assembled these Pezizales sequences with our sequences generated from fungal cultures” Assembled or combined? An assembly represents an alignment of one individual/representative.
Response: We clarify that these sequences were combined in lines 239-241, “We combined these Pezizales sequences with our sequences generated from fungal cultures and aligned them using MAFFT v 7.471 (Katoh and Toh 2010)”.
- Line 221 – “Alignments were manually improved in Se-AL v 2.0a11 (Rambaut 2007).” What do you mean by "improvement"? Did you trim a part of the alignment that could not be aligned, or something else?
Response: Here we mean that we manually checked the alignment done by the MAFFT program to make sure that the resulting alignment was correct, this is explained in lines 241-242, “Alignments were visualized and examined for accuracy in Se-AL v 2.0a11 (Rambaut 2007).”
- Line 222 - Could you please explain why the phylogenetic tree is necessary for your work? Wouldn't it be sufficient to do a simple comparison with the BLAST database to make sure that your sequence is close to the ascoma you are interested in? Why did you choose the LSU, and not ITS tree? Were the ITS and LSU phylogenies congruent?
Response: The tree is necessary because some sequences have no reference sequences identified at the species level and needed phylogenetic analyses to figure out their statistically supported placements. LSU alignments are commonly used to compare sequences across genera. This is made clear in the Figure legend, lines 261-262, “One of the most likely RAxML phylogenetic trees estimated from an LSU alignment showing the wide diversity of single-spore isolates among other Pezizales fungi.”
- Line 279, Figure 4 - It seems the figure is not correctly oriented. Please correct.
Response: Thank you, this has been fixed.
- Lines 293-301 - As mentioned at the beginning, the process described in the manuscript is quite complex and requires special preparation. Please either provide more details about how you plan to use the citizen scientists' data (e.g. additional photos, location data, etc.), or put less emphasis on the citizen science and more on the scientific benefit of your research.
Response: We now add details of a citizen scientist's role each time they are mentioned.
Lines 28-30 - “This method is sufficiently straightforward that citizen scientists from distant locations could use this approach to capture spores and subsequently mail them with voucher specimens to a research lab for further study. ”
Lines 90-91 - “This accessible method enables collection of ascospores in the field, by citizen scientists and researchers alike, so that researchers may generate cultures at a later date.”
Lines 410-413 - “Researchers can provide collection materials (sterile petri plates, tape, some simple instructions, and perhaps a pre paid, padded envelope addressed to the research lab) to citizen scientists who can then make spore collections available for future research. ”